# Association of Inflammatory and Metabolic Biomarkers with Mitral Annular Calcification in Type 2 Diabetes Patients

**DOI:** 10.3390/jpm12091484

**Published:** 2022-09-10

**Authors:** Elena-Daniela Grigorescu, Cristina-Mihaela Lăcătușu, Mariana Floria, Georgiana-Diana Cazac, Alina Onofriescu, Alexandr Ceasovschih, Ioana Crețu, Bogdan-Mircea Mihai, Laurențiu Șorodoc

**Affiliations:** 1Unit of Diabetes, Nutrition and Metabolic Diseases, “Grigore T. Popa” University of Medicine and Pharmacy, 700115 Iași, Romania; 2Clinical Center of Diabetes, Nutrition and Metabolic Diseases, “St. Spiridon” County Clinical Emergency Hospital, 700111 Iași, Romania; 3Internal Medicine, “Grigore T. Popa” University of Medicine and Pharmacy, 700115 Iași, Romania; 4Department of Internal Medicine, “St. Spiridon” County Clinical Emergency Hospital, 700111 Iași, Romania; 5Crețu R. Ioana PFA, 707020 Aroneanu, Romania

**Keywords:** mitral annular calcification, type 2 diabetes mellitus, TNF-α, HOMA-IR, C-peptide, hsCRP

## Abstract

(1) Background: Type 2 diabetes mellitus (T2DM) contributes to cardiovascular disease and related mortality through the insidious effects of insulin resistance and chronic inflammation. Mitral annular calcification (MAC) is one such degenerative process promoted by T2DM. (2) Methods: This is a post hoc analysis of insulin resistance, inflammation, and hepatic steatosis markers in T2DM patients without atherosclerotic manifestations, but with incidental echocardiographic detection of mild MAC. (3) Results: 138 consenting patients were 49.3% men, 57.86 years old, with a history of T2DM of 6.16 years and HbA_1c_ 8.06%, of whom sixty had mild MAC (43.47%). The statistically significant differences between patients with/without MAC were higher HOMA C-peptide and C-peptide index for insulin resistance, higher TNF-α for inflammation, and lower estimated glomerular filtration rate. High-sensitive C-reactive protein (hsCRP) was significantly associated with insulin resistance and the strength of the relationship was higher in the MAC group. Predictive of MAC were TNF-α, HOMA C-peptide, and especially hepatic steatosis and hypertension. (4) Conclusions: MAC was more prevalent than reported in the literature. Insulin resistance and inflammation were predictive of MAC, but significant markers differ across studies. Widely available routine tests and echocardiographic assessments are useful in the early identification of mitral annular calcifications in diabetes patients.

## 1. Introduction

Mitral annular calcification (MAC) is a chronic and degenerative process leading to the precipitation of calcium and phosphate in the fibrous structure surrounding the mitral valve leaflets. It is also more than that, as MAC has been associated with various anomalies regarding lipid and mineral metabolisms, chronic kidney disease, and inflammation [1]. The known risk factors for MAC development are advancing age, female sex, obesity, hypertension, left ventricular hypertrophy, dyslipidemia, diabetes mellitus, advanced chronic kidney disease, osteoporosis, and smoking [2,3].

In turn, MAC is a marker for multi-site atherosclerotic plaques affecting coronary and peripheral arteries. As MAC is more prevalent in the elderly, an association with mitral valve diseases (e.g., mitral stenosis) is also to be expected. All these conditions have emerged as important causes of cardiovascular disease and mortality, arrhythmias (atrial fibrillation), and complications related to mitral valve surgery [4,5,6,7,8].

The most common and conclusive diagnostic approach is echocardiographic: MAC appears as an irregular and echo-dense structure, the extent of which defines the severity of the calcification. In mild and moderate cases, the echo density does not exceed 180° and 180–270°, respectively, while calcium deposits extending beyond 270° are considered severe [9,10]. Computer tomography provides a more accurate, detailed view of the calcification within the mitral valve apparatus and helps distinguish MAC from aortic valve and coronary calcium, but it is also more costly and less widely available [11].

The risk factors for heart valve calcifications—MAC and aortic valve sclerosis (AVS)—have received increasing scientific attention in recent years. Both type 2 diabetes mellitus and non-alcoholic fatty liver disease (NAFLD) have been independently associated with the incidence of MAC [6,12]. Moreover, the contribution of underlying metabolic conditions to micro- and macrovascular complications, adverse cardiovascular events, and to related deaths is now an established fact [12,13,14]. Before manifesting symptomatically and becoming life threatening, heart valve calcification starts as a slow and silent process, so researchers are now considering the subclinical pathogenic mechanisms that are especially active and mutually enhancing in the presence of diabetes [15]. It is hard to tell if insulin resistance is cause or effect, as it seems to operate as a “two-way street” with ramifications beyond the immediately obvious glycemic imbalance typical of diabetes [16]. Insulin resistance is commonly evaluated using the Homeostatic Model Assessment of Insulin Resistance (HOMA-IR) and other similar formulae considering the level of C-peptide [17].

Insulin resistance and chronic inflammation amplify each other and have a more potent detrimental effect on many of the body’s normal processes. By initiating the progressive reduction in mass and function of beta-cells, for instance, they undermine the body’s adaptive immunity [13,14]. Inflammation is assessed by standardized methods based on the sampling of certain proinflammatory cytokines, of which the most common in clinical practice is the high-sensitivity C-reactive protein (hsCRP) from the pentraxin protein family. Normally synthesized by the liver, hsCRP is one of the clearest indicators of acute inflammation and, as such, it is considered in cardiovascular risk assessment [18]. The fact that smooth muscle cell lymphocytes and monocytes in atherosclerotic lesions can also produce hsCRP makes it a sensitive marker for chronic inflammation and emerging atherosclerotic disease as well [19,20].

Two mediators of hsCRP are equally useful in evaluating chronic, subclinical, low-grade inflammatory status: interleukin 6 (IL-6) and TNF-α. Interleukin 6 is synthesized when inflammation occurs, and it stimulates hsCRP production. It also contributes to the activation, growth, and differentiation of B and T cells, which are essential to the body’s immune system. Elevated blood levels of IL-6 have been moderately associated with diabetes and dyslipidemia, and significant predictive value has been shown for myocardial infarction and death related to coronary artery disease [21,22]. Moreover, hsCRP and IL-6 have been independently associated with the incidence of MAC, proof that inflammation plays a significant role in the pathogenesis and progression of MAC [1]. The other, TNF-α, is known for wide-ranging effects, among which are the promotion of insulin resistance and of metabolic processes that release the energy necessary for inflammatory reactions to occur at the cellular level [23].

Non-alcoholic fatty liver disease (NAFLD) is a multisystemic spectrum of diseases starting from simple steatosis and non-alcoholic steatohepatitis (NASH), progressing to fibrosis, cirrhosis, or even hepatocellular carcinoma. With a prevalence of over 25% of the global adult population, NAFLD often develops as a hepatic manifestation of metabolic syndrome [24,25]. NAFLD is closely related to obesity, insulin resistance, dyslipidemia, type 2 diabetes, and hypertension. More than 50% of people with type 2 diabetes (T2DM) and 90% of morbidly obese people have NAFLD [26]. Recent data also linked NAFLD to an increased prevalence and incidence of atherosclerosis diseases due to shared risk factors [27]. The high-risk morbidity and mortality of NAFLD are related to its extrahepatic manifestations and comorbidities, primarily cardiovascular [12,28,29]. NAFLD is a risk factor for valvular cardiac calcifications such as aortic valve sclerosis (AVS) and mitral annulus calcification (MAC), as well as for cardiomyopathy, cardiac arrhythmias, ectopic fat deposit, augmenting the risk of cardiovascular events [30]. In two noteworthy studies on patients with diabetes, NAFLD was independently associated with aortic valve sclerosis and mitral annulus calcification, after adjusting for other metabolic factors [31,32].

At present, the early diagnosis of NAFLD is an important objective of non-invasive, risk-free, imagistic assessment before proceeding to invasive investigations such as the liver biopsy, the current “gold standard” [24]. Clinical scores and serum biomarkers may also be included to assess the degree of steatosis, detect the presence or extent of a tissue lesion characteristic of NASH, and identify and quantify liver fibrosis in correlation with imaging data. Fatty Liver Index (FLI), the NAFLD Liver Fat Score (LFS), and the Hepatic Steatosis Index (HSI) are three of the validated scores based on clinical and biological parameters for the diagnosis of hepatic steatosis, commonly used in epidemiological NAFLD studies [25,33]. Recently, the Triglyceride Glucose Index (TyG) and other TyG-driven parameters have been proposed as new markers for detecting fatty liver in patients with diabetes mellitus. The TyG index is already as an insulin resistance marker linked with hepatic steatosis in patients with T2DM [34].

The aim of this study was to assess the hypothesized value of inflammation, insulin resistance, and hepatic steatosis markers as screening tools for the early diagnosis of mitral annular calcification in patients with type 2 diabetes without atherosclerotic manifestations.

## 2. Materials and Methods

### 2.1. Patient Enrolment, Inclusion, and Exclusion Criteria

This is a post hoc analysis of cross-sectional data from a larger, prospective study conducted at the Clinical Centre of Diabetes, Nutrition, and Metabolic Diseases Iași between June 2016 and February 2018. The data reported and discussed in this article were collected upon the patients’ enrolment. All the patients consented in writing. The study was conducted according to the guidelines of the Declaration of Helsinki and approved by the Ethics Committee of “Grigore T. Popa” University of Medicine and Pharmacy Iași, and by the Ethics Committee of the “St. Spiridon” Emergency Clinical Hospital also in Iași (no. 63274/16.12.2015).

Apart from written consent and the diagnosis of type 2 diabetes, the other key inclusion criterion was poor glycemic control, defined as HbA_1c_ levels > 7%. According to national protocols at the time, exceeding this threshold made patients treated with metformin and/or sulfonylurea or acarbose eligible for additional incretin-based medication as part of their state-funded medical insurance.

We considered ineligible for this study, patients with any acute complications of diabetes. Patients with atherosclerotic cardiovascular disease or valvular heart disease, moderate/severe mitral annular calcification, dysrhythmias, or cardiac pacemakers were excluded from the analysis, as were patients with past medical histories of inflammatory and severe conditions, either acute or chronic (pancreatitis, liver failure/viral hepatitis, gastrointestinal and kidney diseases, malignancies). Smoking was also a criterion for exclusion.

### 2.2. Clinical Investigations and Data Collection

The metabolic profile and inflammatory status of the patients were assessed based on routine and immunological bloodwork, liver, and renal function tests (total cholesterol, HDL-cholesterol, LDL-cholesterol, triglycerides, glycemia, HbA_1c_, uric acid, insulin, C-peptide, high-sensitive CRP, TNF-α, IL-6, alanine amino transferase (ALT), alanine aspartate transferase (AST), gamma-glutamyl transferase (GGT), creatinine, and estimated glomerular filtration rate (eGFR) by CKD-EPI formula, the urine albumin/creatinine ratio).

Clinically validated formulae were used to calculate insulin resistance indices [35]:HOMA-IR = (fasting glycemia in mg/dL × insulinemia in μU/mL)/405;HOMA C-peptide = (fasting glycemia in mg/dL/18 × C-peptide in ng/mL × 3.003)/22.5;Index C-peptide = 20/[(C-peptide in ng/mL × 3003) × (fasting glycemia (mg/dL)/18)].

The Fatty Liver Index (FLI), the Hepatic Steatosis Index (HSI), and the Non-Alcoholic Fatty Liver Disease–Liver Fat Score (NAFLD-LFS) were computed with the e-tool MDCalc to predict the liver steatosis:The FLI formula takes into consideration the body mass index (BMI = body weight in kg/height in m^2^), waist circumference, triglycerides, and gamma-glutamyl transferase (GGT). Scores upwards of sixty are indicative of fatty liver. The exact formula is FLI = (e0.953*loge (triglycerides) + 0.139*BMI + 0.718*loge (GGT) + 0.053*waist circumference − 15.745)/(1 + e0.953*loge (triglycerides) + 0.139*BMI + 0.718*loge (GGT) + 0.053*waist circumference − 15.745) × 100 [36].The HSI score is calculated using the formula HSI = 8 × ALT/AST + BMI (+ 2 if type 2 diabetes yes, + 2 if female). A result of 36 or higher suggests the presence of NAFLD, so clinicians can use HSI to decide if an ultrasound investigation is needed [37].The NAFLD Liver Fat Score is assessed based on the presence of metabolic syndrome, type 2 diabetes, fasting serum insulin, fasting serum AST, and the AST/ALT ratio. A score higher than −0.64 suggests the presence of the disease. The exact formula is NAFLD-LFS = −2.89 + 1.18 × Metabolic Syndrome (Yes: 1, No: 0) + 0.45 × Type 2 Diabetes (Yes: 2, No: 0) + 0.15 × Insulin in µU/L + 0.04 × AST in U/L − 0.94 × AST/ALT [38].

The BARD score was also calculated to predict hepatic fibrosis. BARD works as a system of assigning points depending on the AST/ALT ratio (2 points if ≥0.8), BMI (one point if ≥28 kg/m^2^), and the presence of diabetes (one point). Any combination of two or more points is suggestive of advanced hepatic fibrosis [39].

To assess the risk of NAFLD and gain additional indications of insulin resistance, the TyG index was calculated as the natural logarithm of the level of triglycerides (mg/dL) multiplied by half the value of fasting glucose (mg/dL). The TyG index was then multiplied by the body mass index and by the waist circumference to calculate the other related indices TyG-BMI and TyG-WC, respectively [40,41,42].

The patients’ heart function and the presence of calcifications (MAC and aortic atheromatosis) were evaluated by transthoracic echocardiography in accordance with the newest recommendations, using a Sonoscape SSI 5000 machine [43]. Diastolic dysfunction was assessed by means of bi-dimensional, pulsed wave Doppler echocardiography and Tissue Doppler Imaging, based on the algorithm by which diastolic dysfunction is present when at least half of the following conditions are noted: septal e’ velocity < 7 cm/s, lateral e’ velocity < 10 cm/s, E/e’ > 14, indexed left atrium volume to body surface >34 mL/m^2^, tricuspid regurgitation velocity > 2.8 m/s [44]. In addition, 12-lead electrocardiograms were performed.

Last but not least, the study criteria and the collected data allowed for the calculation of the thromboembolic risk score CHA_2_DS_2_-VASc, which features diabetes as one of the contributing factors, along with congestive heart failure, hypertension, age, stroke, vascular disease, and female sex [45].

### 2.3. Statistical Analysis

The database was compiled and processed using IBM SPSS Statistics for Windows (version 17, SPSS Inc., Chicago, IL, USA). Descriptive statistics included frequencies, central tendencies (means, medians), and variability (standard deviation, minimum and maximum values, and interquartile ranges). Specificity and sensitivity were assessed by plotting the ROC curves and calculating the areas under the curves (AUCs). Parametric and non-parametric tests (*t*-test, Mann–Whitney U) were used to compare means or medians between specific groups (with or without MAC). Significant associations between variables with or without normal distribution were identified using the Pearson and Spearman coefficients, respectively. Univariate models for variables such as the patients’ levels of studied markers, glycated hemoglobin, blood pressure, etc., were analyzed to see which clinical characteristics could predict the risk of MAC. The threshold for statistical significance was set at *p* < 0.05.

## 3. Results

### 3.1. General Clinical Patient Information

The research criteria were met by 138 patients with poorly controlled type 2 diabetes, of whom 68 were men (49.3%) and 70 women (51.7%). The mean age was 57.86 ± 8.82, and 77 patients (55.8%) were between 50 and 64 years old. The general clinical characteristics are summarized below and in Table 1, where these data are presented both overall and grouped based on the presence/absence of mitral annular calcification, the focus of the study.

Excess weight, especially in the abdominal area, was a common general characteristic. The mean value of the patients’ body mass indices was 32.65 ± 5.50 kg/m^2^, well above the threshold for clinical obesity (BMI ≥ 30 kg/m^2^). In fact, only 47 patients had BMI values < 30 kg/m^2^, and waist circumference was abnormal in all but three cases.

The diagnosis of type 2 diabetes had been given an average of 6.16 ± 4.73 years before the study, and 36 patients (26.1%) had been living with the disease for more than 10 years. At the time of enrolment, all the patients’ HbA_1c_ levels exceeded 7%, and two-thirds had levels higher than 7.5%, resulting in an overall, mean value of 8.06% ± 0.99.

In addition, 94 patients (68.1%) had elevated levels of triglycerides > 150 mg/dL. Although less than half were hypercholesterolemic (61 patients, 44.2%), a vast majority had levels of LDL-cholesterol > 70 mg/dL (114 patients, 82.6%). The level of uric acid in the blood was above normal in a third of the cases.

Regarding diabetes-related complications, 58 patients (42%) manifested sensorimotor peripheral neuropathy and ten patients (7.2%) had mild non-proliferative retinopathy. Only twelve patients did not present with comorbidities, while 104 suffered from non-alcoholic steatohepatitis, 99 from dyslipidemia, and 93 from therapy-controlled arterial hypertension.

Regarding insulin resistance based on established HOMA-IR threshold values, half of the patients were highly resistant (>5) and only 21 patients (15.2%) were sensitive to insulin (<2). The statistical significance of the areas under the curve (AUCs) for FLI, HSI, and NAFLD-LFS was very high, pointing to their diagnostic indication of insulin resistance (HOMA IR > 5). The same can be said of the inflammatory markers hsCRP and IL-6, which had significant predictive value for insulin resistance—see Table 2.

Approximately 90% of all samples reached BARD, FLI, and HSI values indicative of hepatic steatosis. This was even more predominant in the case of NAFLD-LFS, for which only seven patients scored below −0.64.

In addition, 72% of the patients had hsCRP levels > 3 mg/L. Approximately 94% of the patients displayed values higher than 4 ng/mL for TNF-α, and approximately 55% of the participants recorded IL-6 values above 2 ng/mL.

The echocardiographic assessment of diastolic function in patients with normal EF (FEVS = 67.14%) showed that only 29% of the patients had normal diastolic function. Determining the degree of severity was possible for 83 patients, while for 15 participants the degree was defined as “indeterminate”. Of the patients who could be diagnosed with diastolic dysfunction (DD), 71 patients (51.4%) had first-degree DD, 5 patients had 2nd degree DD, and another 7 had third-degree DD. Other mean values calculated based on echocardiographic data were:Left ventricular ejection fraction (LVEF) = 67.14 ± 9.35%;Interventricular septal thickness (IVS) = 11.40 ± 1.7 mm;E/A = 1.09 ± 0.46 and E/e’ = 6.54 ± 1.84;Early diastolic filling time (EDT) = 192.88 ± 42.76 ms;Isovolumic relaxation time (IVRT) = 104.23 ± 18.74 ms;Left atrial volume indexed (LAVi) = 43.79 ± 11.84 mL/m^2^.

### 3.2. Evidence of MAC and Clinical Characteristics Based on MAC Status

According to the ultrasound investigations, mitral annular calcification was present in sixty cases (43.47%)—27 male and 33 female patients. All were mild cases, meaning that the extent of focal calcifications was <180° (see Figure 1). Moderate to severe MAC would have required a different assessment approach for diastolic function.

The patient characteristics grouped by MAC status are summarized in Table 1. The results show that, on average, patients with MAC were older, and had higher BMI values and greater abdominal circumference. These differences were not statistically significant, although, clinical obesity was noticeably more common in this group—44 cases (73.3%).

Patients with MAC had higher levels of triglycerides (>150 mg/dL in 65% of cases, and >200 mg/dL in 43% of cases), LDL-cholesterol (>70 mg/dL in 80% of cases), and uric acid (but elevated in only 28% of cases); these, too, were not statistically significant. The estimated glomerular filtration rates (eGFR) were significantly lower (*p* = 0.009), and the urine albumin-to-creatinine ratio (ACR) values were also lower, but not significantly.

Interestingly, although their history of diabetes was shorter, patients with MAC presented with poorer glycemic control, and in 40% of cases, glycated hemoglobin levels exceeded 8%. Higher levels of insulin and insulin resistance markers were also noted, as well as lower values of the C-peptide index; of these, the significant differences were for HOMA C-peptide (*p* = 0.028) and C-peptide index (*p* = 0.032).

Regarding the inflammation markers, the results were mixed: patients with MAC had significantly higher levels of TNF-α (*p* = 0.037), but not significantly lower levels of hsCRP and IL-6.

### 3.3. Associations between Inflammation, Insulin Resistance, and Hepatic Steatosis Markers

Our inferential analysis involving the three sets of markers (inflammation, insulin resistance, and hepatic steatosis) included a view of the entire study cohort as well as a more detailed look at the subgroups of patients with and without MAC. The intensities of the studied associations and their corresponding *p* values are summarized in Table 3.

Overall, of the inflammation markers, hsCRP was significantly associated with all the insulin resistance markers, with weak to moderate strength (see Table 4). The relationship was linear among all except the peptide-C index. Of the steatosis markers, HS index and NAFLD-LFS were significantly associated with all the inflammation markers. Of the insulin resistance markers, HOMA-IR, HOMA C-peptide, TyGiBMI, and TyGiWC were in a significant linear relationship with FLI, HS index, and NAFLD-LFS, while the index C-peptide correlated negatively (see Table 5).

Further analysis based on the presence/absence of MAC revealed a more nuanced picture. For instance, when looking specifically at the subgroup of patients with MAC, the intensity of the significant associations involving the inflammation marker hsCRP increased across the board.

Of all the studied biomarkers, the ROC curves for TNF-alpha (AUC = 0.625, CI 0.531–0.720, *p* = 0.012) and HOMA C-peptide (AUC = 0.605, CI 0.509–0.702, *p* = 0.034) were predictive of MAC presence. These can be seen in Figure 2.

A 1% increase in TNF-α values was associated with 12.5% higher odds of MAC presence, and the critical value of TNF-α 9.53 or higher was found to be predictive of MAC in the respective cases (28 patients). Similarly, a 1% increase in HOMA C-peptide value led to 20.7% higher odds of MAC, and the critical value of 6.42 or higher predicted the presence of MAC in the respective cases (19 patients)—see Table 6.

Additionally, hypertension and hepatic steatosis were each significantly associated with more than a twofold higher risk of MAC (OR 2.16 and 2.37, respectively). The other studied markers, the patients’ sex, and other known comorbidities (including neuropathy) did not achieve significant predictive value for MAC.

## 4. Discussion

This was a prospective study of inflammation, insulin resistance, hepatic steatosis markers, and the presence of mitral annular calcifications in poorly controlled type 2 diabetes patients without manifest atherosclerotic complications. The analysis was based on data from 138 consenting patients with an average history of diabetes of 6.16 years, treated with one or more standard drugs (metformin, sulfonylurea, and acarbose), but whose glycated hemoglobin levels were nevertheless higher than 7%, defined as poorly controlled. The patients were both men and women in almost equal proportions, 57.86 years old on average, clinically obese, with excess abdominal adiposity, and elevated levels of triglycerides and LDL-cholesterol. Non-alcoholic steatohepatitis, dyslipidemia, and hypertension were comorbidities affecting most patients. The blood tests showed high insulin resistance, presence of hepatic steatosis, and inflammation levels indicative of high risk for cardiovascular disease.

The echocardiographic examinations revealed that 98 patients (71%) had some degree of diastolic dysfunction, and 60 patients had mild mitral annular calcifications (43.47%). Regarding diastolic dysfunction, this result is similar to that of another study from 2013, when asymptomatic diastolic dysfunction was found in 68% of 386 patients with diabetes and no cardiovascular diseases. In the same study, however, 28% of patients also had MAC, fewer than in our study. Moreover, MAC was significantly more present in patients with diastolic dysfunction (32%) than without, while in our study the echocardiographic status did not differ significantly between patients with or without MAC [46]. This is an important result of our study. It is well known that chronic diastolic dysfunction reflects left atrial remodeling and, subsequently, a worse prognosis. In addition, the preponderance of non-alcoholic hepatic steatosis in our study points to a high level of risk for cardiovascular disease. Therefore, patients with MAC might have a worse prognosis despite their echocardiographic status not being more severe.

In fact, the prevalence of MAC in our study appears to be substantially higher relative to other published results, though we are aware that the diagnostic approach can lead to different outcomes ranging between 5 and 42% according to a review from 2020 [1]. For example, in one study on patients with diabetes, MAC was found in 28% of cases, while in another study enrolling patients with both diabetes and NAFLD, the reported prevalence was 19.4% [32,46]. Larger-scale research on general and patient populations with a wider range of pathologies points to an 8–15% prevalence of MAC, and advanced age was shown repeatedly to play a significant role [9,47,48]. The fact that more than half of the patients who participated in our study were under the age of 65 makes it noteworthy that mild MAC was found in as many as 43.47% of the participants. This suggests that the early onset of MAC may be a more pervasive phenomenon than previously thought in relation to active patients whose diabetes is being managed with non-insulin medication.

This brings us to another interesting discussion point about the clinical implications of such a high proportion of mild MAC cases in our study. Ever since the 1980s, the concomitant occurrence of MAC and aortic atherosclerosis has been noted in patients both with and without diabetes, prompting the view that MAC is one manifestation of atherosclerosis [6,49,50,51]. All the patients enrolled in our study were asymptomatic in terms of atherosclerosis, and only sixteen presented with aortic atheromatosis, which is the precursor of atherosclerosis. Given the evidence in the literature linking MAC with coronary and systemic atherosclerosis, the presence of mild MAC in 43.47% of patients from our study group can be considered an indication of early, subclinical atherosclerotic pathology and a red flag for all the ensuing risks.

The patients’ general and clinical characteristics analyzed based on the presence or absence of these calcifications invite further discussion. The patients with MAC had shorter histories of diabetes but poorer glycemic control. At the same time, their lipid profiles did not differ significantly. The only statistically significant difference was lower eGFR among patients with MAC. A relationship between MAC and chronic kidney disease has already been shown in the literature; it appears that impaired renal function, such as altered calcium and phosphate metabolism, contributes to MAC development and aggravation [9,52,53]. In a recent study focusing exclusively on patients with severe MAC, the eGFRs were an independent predictor of MAC and, also, the filtration rates of patients who also had diabetes were significantly poorer [48]. Our study excluded patients with overt kidney disease from the beginning and only found mild cases of MAC, yet the glomerular filtration rates of the patients with MAC were already significantly lower. We believe that this result points to a subtle reduction in renal function, which, if left unaddressed, could develop into full-blown chronic kidney disease as well as accelerate the mitral annular calcification process.

Regarding the studied sets of markers, of all the noted differences, the statistically significant ones were higher insulin resistance expressed as HOMA C-peptide, C-peptide index (but not HOMA-IR), and higher inflammatory status expressed as TNF-α (but not IL-6 and hsCRP) in the patients with MAC compared to those without MAC. In our review of the relevant literature, we identified one other study reporting a significantly higher level of TNF-α based on the MAC status of eighty patients who had both type 2 diabetes and stage 2–4 chronic kidney disease. Another similarity to our study is that TNF-α was an associated risk factor for MAC with an adjusted odds ratio of 1.193 at *p* = 0.029; in our study, the odds ratio for TNF-α was 1.125 at *p* = 0.026 [54].

The inferential analysis for the entire group of 138 patients revealed statistically significant associations between the inflammation marker hsCRP and all the insulin resistance markers. The bidirectional relationship between inflammation and insulin resistance has been investigated in studies on diabetes with/without associated metabolic syndrome. Significant positive associations have been found between HOMA-IR and hsCRP in patients with type 2 diabetes across multiple studies [55]. Elevated hsCRP has also been proven predictive of insulin resistance in non-diabetic research participants, helping identify patients with prediabetes and asymptomatic diabetes [56].

Notably, the intensity of all the significant associations involving the inflammation marker hsCRP was higher in our data set describing the patients with MAC. According to the literature, in the presence of MAC, hsCRP was significantly higher in patients with diabetes, even when it did not necessarily yield significant predictive value for MAC (while age, hypertension, and coronary artery disease did) [57]. In our study, it was hypertension and hepatic steatosis that were significantly associated with the highest risk of MAC, followed by HOMA C-peptide and of TNF-α, which were also predictive of MAC presence; a 1% increase in either was associated with substantially higher odds of MAC presence (20.7% and 12.5%, respectively).

Furthermore, most insulin resistance markers were in a significant positive association with the hepatic steatosis markers, adding to existing reports in the literature such as linking HOMA-IR with HSI and FLI [58]. Concurrently, the HS index and NAFLD-LFS were significantly associated with all the inflammation markers, which points to a vicious cycle of mutually enhancing pathological processes that accelerate disease, complication, and risk. In one study on 247 diabetes patients without atherosclerosis (as in our study), the presence of NAFLD, elevated HbA_1c_, and low eGFR were all significant predictive factors for MAC, as well as for both heart valves being affected (MAC and AVS) [31].

It may seem superfluous to discuss other differences between the patients with and without MAC that were not statistically significant, such as their thromboembolic risk scores CHA_2_DS_2_-VASc (3 vs. 2). However, we believe this result further reinforces the point that clinicians should consider their patients’ data as comprehensively as afforded by the tests, investigations, and validated formulae available to them. Patients with mitral annular calcifications develop arrhythmias and, as such, are at risk of thromboembolism [3]; for example, in a study on 388 embolic patients, MAC was identified as the cause in 27% of cases [59]. Moreover, recent findings suggest that MAC is a risk factor for thromboembolism even in patients who, like ours, do not manifest atrial fibrillation [60,61]. To prevent adverse embolic events, diabetes patients incidentally identified with MAC might be considered eligible for treatment typically prescribed in atrial fibrillation—the hypothesized benefits of such an approach could make the object of scientific scrutiny in future trials.

Despite the scientific attention given to all the diabetes-related manifestations, comorbidities, and complications, certain mechanisms and the relationships of causality can be hard to untangle. Inflammation does seem to be the key link in the downward spiral, but the data are inconsistent across different studies. The fact that only some, but not all, markers in any of the studied sets have yielded statistically significant associations with MAC status invites caution. For instance, in a multi-ethnic study on 5895 patients, of whom 789 with diabetes, and who were monitored for two years and a half, the prediction factors for MAC were diabetes and hypertension (like in our study), but also age and interleukin-6, which were not significantly predictive according to our data [62].

The study design and results can inform several directions of further research. For one, a longitudinal monitoring of the same patients, as in the aforementioned study, would yield interesting results about the progression of the calcifications initially identified as mild and any related complications, as well as the onset of MAC in patients who did not initially show echocardiographic evidence of it. This or a version of the study could be expanded to include a larger number of T2DM patients from multiple centers, and the full spectrum of MAC severity. Moreover, as previously suggested, the hypothesized benefits of antihyperglycemic and antithrombotic medication could be assessed in relation to chronic inflammatory status and silent MAC progression. Based on the latest research, both innate and adaptive immune cells are currently accepted to initiate and maintain the complex processes involved in cardiac valve calcification [63,64,65]. As pathogenesis is similar in mitral and aortic valvular calcifications, further research may focus on associations between cellular immunity and early MAC. Existing data in this direction are limited to a few findings showing associations between MAC and neutrophil/lymphocyte ratio, total lymphocyte count, platelet/lymphocyte ratio, and monocyte/high-density lipoprotein ratio [66,67,68,69].

For clinical practice, the study reinforces the importance of diabetologists conducting a competent, comprehensive assessment of their patients’ data available through routine tests and investigations. Insulin resistance, elevated inflammation markers, decreased glomerular filtration rates, and incidental echocardiographic evidence of incipient mitral annular calcification are important clues that atherosclerotic disease is underway and preventable adverse cardiovascular events are looming, even if other telltale signs and symptoms may be absent [15].

### Limitations

The patients’ eligibility for additional, incretin-based medication through state-funded insurance at the time of the study was a factor in our definition of the inclusion criteria. We were specifically interested in patients with type 2 diabetes whose standard treatment was ineffective in maintaining adequate glycemic control, and who would thus benefit from access to the newly available drugs before considering insulin-based treatment.

Methodologically, the post hoc observational nature of the study does not explain cause and effect. Moreover, as a single center, sample size was a challenge but the fact that the data came from consecutive patients adds to the statistical robustness of the results. We refrained from reporting on the multivariate analysis due to small patient numbers.

CT scans are more accurate in detecting early MAC. However, due to limited diagnostic resources, our study used echocardiography. The echocardiographic detection of early MAC can prompt the planning of subsequent echocardiographic investigations and, when available or more imperatively necessary, CT scans.

## 5. Conclusions

Mitral annular calcification is a slow, silent process that eventually degenerates into overt cardiovascular disease, adverse events, and related mortality. In patients with type 2 diabetes mellitus and non-alcoholic fatty liver disease, the converging effects of insulin resistance, chronic inflammation, and hepatic steatosis undermine renal function and the body’s calcium metabolism, facilitating the emergence of calcium deposits in wrong places such as heart valves.

Our study shows that widely available, affordable, routine blood tests, and echocardiographic investigations can provide the necessary data for the comprehensive assessment of a diabetes patient at risk of developing MAC. Importantly, we found incidental echocardiographic evidence that mild MAC can be more prevalent in type 2 diabetes patients and at a relatively younger age than previously reported in the literature.

Statistically significant associations were established between inflammation and insulin resistance across the studied group, but they were stronger in patients with MAC, who also had significantly lower glomerular filtration rates. The studied markers predictive of MAC were TNF-α for inflammation and HOMA C-peptide for insulin resistance, and the computed risk for MAC was highest in the presence of hepatic steatosis and hypertension.

## Figures and Tables

**Figure 1 jpm-12-01484-f001:**
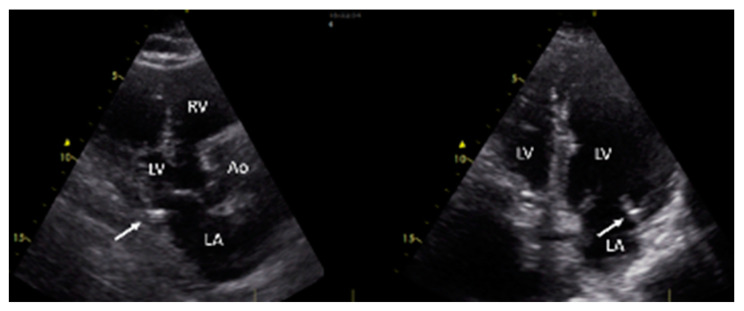
Transthoracic echocardiography in long parasternal view and modified apical four-chamber view showing mitral annular calcification in an enrolled patient. Ao—aorta; LA—left atrium; LV—left ventricle; RV—right ventricle.

**Figure 2 jpm-12-01484-f002:**
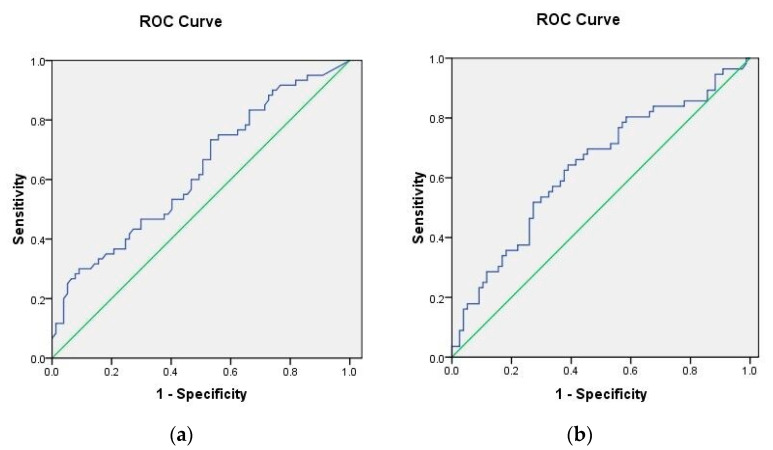
Receiver operating characteristic (ROC) curves for (**a**) TNF-α and for (**b**) HOMA C-peptide.

**Table 1 jpm-12-01484-t001:** Patient characteristics overall as well as by MAC status.

Studied Variable	Overall (N = 138)	With MAC (N = 60)	Without MAC (N = 78)	*p*-Value
**General characteristics**				
Age (years)	57.86 ± 8.82	59.23 ± 7.37	56.79 ± 9.69	0.09
Sex (male %)	49.30	45	52.56	0.378
BMI (kg/m^2^)	32.65 ± 5.50	33.28 ± 5.26	32.16 ± 5.67	0.23
Waist circumference (cm)	109.13 ± 10.74	109.68 ± 10.20	108.96 ± 11.22	0.70
**Diabetes-related**				
Diabetes duration * (years)	5 (8)	4.5 (7)	6 (8)	0.36
Neuropathy (%)	44.2	46.7	42.3	0.609
HbA_1c_ * (%)	7.8 (1.11)	7.8 (1.36)	7.8 (1)	0.55
Fasting glycemia * (mg/dL)	162 (46)	170.50 (40)	160 (52)	0.22
Insulin * (µIU/mL)	11.2 (9.39)	12.5 (10.1)	10.40 (8.49)	0.146
C-peptide * (ng/mL)	3.26 (2.22)	3.38 (2.06)	2.81 (1.46)	0.08
HOMA-IR	5.74 ± 3.87	6.05 ± 3.65	5.50 ± 4.04	0.41
HOMA C-peptide	4.02 ± 2.10	4.48 ± 2.29	3.67 ± 1.89	0.028 **
Index C-peptide *	0.24 (0.19)	0.20 (0.16)	0.27 (0.21)	0.032 **
**Lipid profile**				
Total cholesterol (mg/dL)	195.33 ± 46.11	197.93 ± 51.05	193.33 ± 42.16	0.563
LDL-cholesterol (mg/dL)	103.12 ± 38.96	105.29 ± 42.46	101.46 ± 36.23	0.577
HDL-cholesterol (mg/dL)	56.79 ± 15.27	58.28 ± 16.49	55.65 ± 14.26	0.319
Triglycerides (mg/dL)	202.57 ± 90.46	190.15 ± 89.20	212.12 ± 90.84	0.158
TyGi	9.64 ± 0.52	9.58 ± 0.49	9.68 ± 0.54	0.298
TyGi-BMI	314.75 ± 57.23	305.94 (60)	306.98 (68)	0.414
TyGi-WC	1052.87 ± 118.52	1050.63 ± 109.69	1054.60 ± 125.55	0.846
**Hepatic status**				
ALT * (U/L)	29 (18)	30 (27)	29 (17)	0.667
AST * (U/L)	22 (17)	22 (16)	23 (18)	0.995
GGT * (U/L)	34 (33)	41.5 (35)	32 (33)	0.122
FLI *	87.71 (22)	89.19 (20)	88.39 (24)	0.609
HSI	42.11 ± 6.29	42.70 ± 6.32	41.65 ± 6.27	0.337
NAFLD-LFS	1.51 ± 1.71	1.64 ± 1.51	1.41 ± 1.85	0.43
BARD	2.67 ± 1.05	2.60 ± 0.94	2.73 ± 1.13	0.573
**Kidney function**				
eGFR (mL/min/1.73 m^2^)	82 ± 16.37	78.57 ± 15.12	86.58 ± 19.31	0.009 **
ACR (mg/g)	27.14 ± 48.64	22.15 ± 36.49	31.13 ± 56.54±	0.765
Uric acid (mg/dL)	5.48 ± 1.43	5.54 ± 1.44	5.43 ± 1.43	0.66
**Inflammation markers**				
hsCRP * (mg/L)	5.35 (9.18)	5.11 (8.61)	5.39 (9.11)	0.31
IL-6 (pg/mL)	3.52 ± 4.66	2.83 ± 1.27	3.49 ± 3.08	0.09
TNF-α (pg/mL)	8.73 ± 7.66	9.28 ± 6.18	7.41 ± 3.30	0.037 **
**Comorbidities**				
Dyslipidemia (%)	71.7	78.3	66.7	0.131
Hypertension (%)	67.4	76.7	60.3	0.041 **
Steatohepatitis (%)	75.46	85	70.5	0.046 **
**Cardiac status**				
Diastolic dysfunction (%)	60.14	63.3	54.2	0.385
LV ejection fraction	67.14 ± 9.35	66.37 ± 8.86	67.73 ± 9.72	0.398
FS (%)	38.16 ± 7.89	37.30 ± 7.31	38.85 ± 8.30	0.26
E/A *	0.93 (0.68)	0.84 (0.59)	1.07 (0.71)	0.072
E/e’	6.54	6.28 ± 1.74	6.75 ± 1.90	0.13
EDT *	190 (58)	192.5 (55)	190 (59)	0.216
IVRT *	104 (25)	100 (25)	105 (25)	0.617
IVS *	11 (2)	12 (3)	12 (2)	0.758
LVPW *	11 (2)	11 (2)	11 (2)	0.732
LAVi *	42 (18)	42 (19)	40.5 (17)	0.193
CHA_2_DS_2_-VASc *	3 (1)	3 (1)	2 (1)	0.360
NT-proBNP * (pg/mL)	63 (77.1)	49.23 (83.46)	64 (72.57)	0.961

* Data are expressed as medians and IQR (abnormal distribution). ** Statistical significance. MAC: mitral annular calcification; BMI: body mass index; HbA_1c_: glycated hemoglobin; HOMA-IR: Homeostatic Model Assessment of Insulin Resistance; HOMA C-peptide: Homeostatic Assessment Model of C-peptide; TyGi: Triglyceride Glucose Index; TyGi-BMI: Triglyceride Glucose Index related to body mass index; TyGi-WC: Triglyceride Glucose Index related to waist circumference; ALT: alanine amino transferase; AST: alanine aspartate transferase; GGT: gamma-glutamyl transpeptidase; FLI: fatty liver index score; HSI: Hepatic Steatosis Index score; NAFLD-LFS: Non-Alcoholic Fatty Liver Disease–Liver Fat score; eGFR: estimated glomerular filtration rate; ACR: albumin-to-creatinine ratio; hsCRP: high-sensitive C-reactive protein; IL-6: interleukin 6; TNF-α: tumor necrosis factor-alpha; A: mitral A wave velocity (atrial contraction) with pulsed Doppler; E: mitral E wave velocity (rapid filling) with pulsed Doppler; e’: mitral annular velocity with tissue Doppler imaging; DT: E-wave deceleration time; IVRT: isovolumic relaxation time; IVS: interventricular septum; NT-proBNP: N terminal-brain natriuretic peptide; LV: left ventricular; LVPW: left ventricular posterior wall; LAVi: indexed left atrium volume.

**Table 2 jpm-12-01484-t002:** The predictive value of inflammatory markers and hepatic steatosis markers.

Variables	AUC	Standard Error	*p*-Value	95% CI
hsCRP (mg/L)	0.706	0.045	0.000	0.619–0.794
IL-6 (pg/mL)	0.626	0.048	0.011	0.530–0.721
TNF-α (pg/mL)	0.513	0.050	0.789	0.415–0.612
FLI	0.763	0.040	0.000	0.683–0.842
HSI	0.704	0.045	0.000	0.617–0.792
NAFLD-LFS	0.900	0.026	0.000	0.849–0.952

hsCRP: high-sensitive C-reactive protein; IL-6: interleukin 6; TNF-α: tumor necrosis factor; FLI: Fatty Liver index; HIS: Hepatic Steatosis Index; NAFLD-LFS: Non-Alcoholic Fatty Liver Disease–Liver Fat Score.

**Table 3 jpm-12-01484-t003:** Statistical associations between biomarkers for the two studied groups of diabetes patients without and with mitral annular calcification.

	MAC Status	FLI	NAFLD-LFS	HSI	BARD	TNF-α	IL-6	hsCRP	HOMA-IR	HOMA C Peptide	Index C-Peptide	TyG Index	TyGi BMI	TyGi WC
**FLI**	No	1.000	0.578 **	0.660 **	0.214	−0.059	0.239 *	0.268 *	0.569 **	0.482 **	−0.480 **	0.387 **	0.833 **	0.837 **
	Yes	1.000	0.507 **	0.541 **	0.172	−0.029	0.144	0.326 *	0.369 **	0.353 **	−0.348 **	0.286 *	0.825 **	0.842 **
**NAFLD-LFS**	No	0.578 **	1.000	0.351 **	0.133	0.003	0.289 *	0.347 **	0.768 **	0.526 **	−0.248 *	0.135	0.498 **	0.434 **
	Yes	0.507 **	1.000	0.360 **	0.078	0.040	0.086	0.265 *	0.769 **	0.543 **	−0.568 **	0.189	0.335 **	0.397 **
**HSI**	No	0.660 **	0.351 **	1.000	0.493 **	0.033	0.267 *	0.361 **	0.445 **	0.305 **	−0.305 **	0.132	0.778 **	0.635 **
	Yes	0.541 **	0.360 **	1.000	0.444 **	0.144	0.128	0.236	0.266 *	0.215	−0.217	−0.199	0.744 **	0.552 **
**BARD**	No	0.214	0.133	0.493 **	1.000	−0.006	0.180	0.188	0.102	0.041	−0.040	−0.015	0.239 *	0.123
	Yes	0.172	0.078	0.444 **	1.000	0.303 *	0.175	0.140	0.058	0.028	−0.036	−0.126	0.208	0.129
**TNF-α**	No	−0.059	0.003	0.033	−0.006	1.000	0.297 **	−0.001	−0.010	0.128	−0.127	0.042	−0.001	−0.029
	Yes	−0.029	0.040	0.144	0.303 *	1.000	0.359 **	0.002	−0.032	0.176	−0.182	−0.068	0.037	0.009
**IL-6**	No	0.239 *	0.289 *	0.267 *	0.180	0.297 **	1.000	0.495 **	0.188	0.099	−0.100	−0.037	0.280 *	0.228 *
	Yes	0.144	0.086	0.128	0.175	0.359 **	1.000	0.119	0.159	0.198	−0.199	0.042	0.182	0.164
**hsCRP**	No	0.268 *	0.347 **	0.361 **	0.188	−0.001	0.495 **	1.000	0.317 **	0.196	−0.201	−0.051	0.293 *	0.266 *
	Yes	0.326 *	0.265 *	0.236	0.140	0.002	0.119	1.000	0.354 **	0.313 *	−0.316 *	0.346 **	0.346 **	0.305 *

* Statistically significant associations where *p* < 0.05. ** Statistically significant associations where *p* < 0.001. ALT: alanine amino transferase; AST: alanine aspartate transferase; BARD: BARD score; BMI: body mass index; FLI: Fatty Liver Index score; GGT: gamma-glutamyl transpeptidase; HbA_1c_: glycated hemoglobin; HOMA-IR: Homeostatic Model Assessment of Insulin Resistance; HOMA C-peptide: Homeostatic Assessment Model of C-peptide; hsCRP: high-sensitive C-reactive protein; HSI: Hepatic Steatosis Index score; IL-6: interleukin 6; MAC: mitral annular calcification; NAFLD-LFS: Non-Alcoholic Fatty Liver Disease–Liver Fat score; TNF-α: tumor necrosis factor-alpha; TyGi: Triglyceride Glucose Index; TyGi-BMI: Triglyceride Glucose Index related to body mass index; TyGi-WC: Triglyceride Glucose Index related to waist circumference.

**Table 4 jpm-12-01484-t004:** Significant correlations between hsCRP and insulin resistance markers.

Parameters	Coefficient r	*p*-Value
HOMA-IR	0.331	<0.001
HOMA C-peptide	0.256	0.003
Index C-peptide	−0.259	0.002
TyGi-BMI	0.318	<0.001
TyGi-WC	0.280	0.001

hsCRP: high-sensitive C-reactive protein; HOMA-IR: Homeostatic Model Assessment of Insulin Resistance; HOMA C-peptide: Homeostatic Assessment Model of C-peptide; TyGi-BMI: Triglyceride Glucose Index related to body mass index; TyGi-WC: Triglyceride Glucose Index related to waist circumference.

**Table 5 jpm-12-01484-t005:** Significant correlations between inflammation, insulin resistance, and hepatic steatosis markers.

Parameters	FLI	NAFLD-LFS	HSI
TNF-α	*r* = −0.037, *p* = NS	*r* = 0.04, *p* = NS	*r* = 0.198, *p* = 0.02
IL-6	*r* = 0.198, *p* = 0.02	*r* = −0.254, *p* = 0.003	*r* = 0.067, *p* = NS
hsCRP	*r* = 0.288, *p* = 0.001	*r* = 0.323, *p* < 0.001	*r* = 0.301, *p* < 0.001
HOMA-IR	*r* = 0.482, *p* < 0.001	*r* = 0.777, *p* < 0.001	*r* = 0.371, *p* < 0.001
HOMA C-peptide	*r* = 0.419, *p* < 0.001	*r* = 0.583, *p* < 0.001	*r* = 0.264, *p* = 0.002
Index C-peptide	*r* = −0.418, *p* < 0.001	*r* = −0.582, *p* < 0.001	*r* = −0.267, *p* = 0.002
TyGi-BMI	*r* = 0.826, *p* < 0.001	*r* = 0.476, *p* < 0.001	*r* = 0.773, *p* < 0.001
TyGi-WC	*r* = 0.837, *p* < 0.001	*r* = 0.421, *p* < 0.001	*r* = 0.548, *p* < 0.001

TNF-α: tumor necrosis factor-alpha; IL-6: interleukin 6; hsCRP: high-sensitive C-reactive protein; HOMA-IR: Homeostatic Model Assessment of Insulin Resistance; HOMA C-peptide: Homeostatic Assessment Model of C-peptide; TyGi-BMI: Triglyceride Glucose Index related to body mass index; TyGi-WC: Triglyceride Glucose Index related to waist circumference; FLI: fatty liver index score; HSI: Hepatic Steatosis Index score; NAFLD-LFS: Non-Alcoholic Fatty Liver Disease–Liver Fat score; NS: not significant.

**Table 6 jpm-12-01484-t006:** Univariate analysis models for risk of MAC.

Variables	Coefficient (B)	SE	Exp (B) Odds Ratio	95% CI	*p*-Value
TNF-alpha (pg/mL)	0.118	0.053	1.125	1.014–1.248	0.026 *
IL-6 (pg/mL)	−0.132	0.091	0.876	0.733–1.04	0.145
hsCRP (mg/L)	−0.14	0.019	0.986	0.950–1.02	0.453
HOMA-IR	0.037	0.044	1.037	0.957–1.132	0.409
HOMA-C Peptide	0.188	0.085	1.207	1.021–1.427	0.028 *
Index C–peptide	−0.766	0.689	0.465	0.120–1.792	0.266
NTproBNP (pg/mL)	0.001	0.002	1.001	0.997–1.005	0.609
CHA_2_DS_2_VASc	0.104	0.183	1.110	0.776–1.589	0.568
Age (years)	0.032	0.02	1.033	0.993–1.074	0.109
HbA_1c_ (%)	0.147	0.173	1.158	0.826–1.624	0.395
Hypertension	0.773	0.383	2.167	1.023–4.591	0.04 *
Hepatic steatosis	0.863	0.439	2.370	1.003–5.598	0.04 *
Dyslipidemia	0.592	0.395	1.808	0.834–3.919	0.134

* Statistical significance (*p* < 0.05). CHA_2_DS_2_VASc: thromboembolic risk score; HbA_1c_: glycated hemoglobin; HOMA-IR: Homeostatic Model Assessment of Insulin Resistance; HOMA C-peptide: Homeostatic Assessment Model of C-peptide; hsCRP: high-sensitive C-reactive protein; TNF-α: tumor necrosis factor-alpha.

## Data Availability

Readers interested in finding out more about the enrolment criteria, initial clinical investigations, and sample processing techniques can refer to Grigorescu et al., 2021 [70].

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
