# Peer review of "Association of Inflammatory and Metabolic Biomarkers with Mitral Annular Calcification in Type 2 Diabetes Patients"

_jpm, 2022, doi:10.3390/jpm12091484_

Round 1

Reviewer 1 Report

In this manuscript, Grigorescu et al., analyzed the data from 138 patients with an average history of type 2 diabetes of 6.16 years and found that 43.47% of these patients had mild mitral annular calcification (MAC), which was more prevalent than reported in the literature. The patients with MAC had stronger inflammation, more insulin resistance and significantly lower glomerular filtration rates. TNFa, HOMA C-peptide, and especially hepatic steatosis and hypertension can be used for the prediction of MAC. So widely available routine tests and echocardiographic assessments are useful in the early identification of MAC in diabetes patients. This is a very interesting topic and the data are clearly presented. Their findings have the potential for clinical application.

Author Response

We thank the reviewer for the positive appreciation of our manuscript. Based on the suggestions from other reviewers, we operated some changes that we trust to increase the global clarity and value of the manuscript.

Reviewer 2 Report

I suggest authors should add acute complications of diabetes in exclusion criteria and to shorten the part related to limitations. 

Author Response

We are grateful to the reviewer for these helpful suggestions. We modified the text accordingly, which we think it consistently helped the manuscript to acquire better clarity and logic.

We have indeed considered ineligible for this study all patients with any acute complications of diabetes. We have made this clear in the text by adding a dedicated sentence at the beginning of the exclusion criteria paragraph (lines 142-143).

We have reduced the section about study limitations according to the suggestion of the reviewer (lines 520-547).

Reviewer 3 Report

In this manuscript, the authors investigated the association of inflammatory and metabolic markers with mitral annular calcification (MAC) in type 2 diabetic patients. Medical history of 138 consenting patients with long time T2D was detected, 60 of them also were also diagnosed with MAC. The data shows that there is significant difference in HOMA c-peptide and c and glomerular filtration rate between patient groups with or without MAC.  Moreover, higher c-peptide and TNF alpha were found in MAC patients. And blood test also showed high insulin resistance, presence of hepatic steatosis, and inflammation levels indicated the high risk for cardiovascular disease. This study showed the widely available and affordable blood test and echocardiographic investigation to predict the risk of patients with T2D to develop MAC. The study is interesting and novel and enlarge the knowledge to predict the risk of patients with long time T2D to develop the MAC. There is an issue need to be resolved before this manuscript accept.

Based on data, the inflammation is relative to the development of MAC in T2D patients, what about the immune cells classification at the different stage of the MAC development?

Author Response

We thank the reviewer for this helpful suggestion. Recent research does indeed suggest the involvement of both innate and adaptive immune cells in the initiation and maintenance of complex processes underlying cardiac valve calcification. As mitral annular calcification is concerned, current data is limited to a few findings that show associations between MAC and total lymphocyte count, neutrophil/lymphocyte ratio, platelet/lymphocyte ratio, and monocyte/high-density lipoprotein ratio. Further research focusing on the associations between cellular immunity and early MAC may bring supplementary evidence in this field. We have introduced all these considerations and related references into the manuscript (lines 503-510).